# Thyrotropin Levels in Patients with Coronavirus Disease 2019: Assessment during Hospitalization and in the Medium Term after Discharge

**DOI:** 10.3390/life12122014

**Published:** 2022-12-02

**Authors:** Abdallah Al-Salameh, Noémie Scherman, Imane Adda, Juliette André, Yoann Zerbib, Julien Maizel, Jean-Daniel Lalau, Etienne Brochot, Claire Andrejak, Rachel Desailloud

**Affiliations:** 1Department of Endocrinology, Diabetes Mellitus and Nutrition, Amiens University Hospital, F-80054 Amiens, France; 2PériTox UMR-I 01, University of Picardie Jules Verne, Chemin du Thil, F-80025 Amiens, France; 3Medical Intensive Care Unit, Amiens University Hospital, F-80054 Amiens, France; 4Laboratory of Virology, Amiens University Hospital, F-80054 Amiens, France; 5Department of Pulmonary Diseases, Amiens University Hospital, F-80054 Amiens, France

**Keywords:** coronavirus disease 2019, COVID-19, non-thyroidal illness syndrome, thyroid-stimulating hormone, hypothalamic-pituitary-thyroid axis, thyroid

## Abstract

Background: The objectives of this study were (1) to compare TSH levels between inpatients with critical versus non-critical coronavirus disease 19 (COVID-19), and (2) to describe the status of TSH levels three months after hospitalization. Methods: We collected data on adult patients hospitalized with COVID-19 at Amiens University Hospital. We compared TSH levels between inpatients with critical (intensive care unit admission and/or death) versus non-critical COVID-19. Thereafter, survivors were invited to return for a three-month post-discharge visit where thyroid function tests were performed, regardless of the availability of TSH measurement during hospitalization. Results: Among 448 inpatients with COVID-19, TSH assay data during hospitalization were available for 139 patients without prior thyroid disease. Patients with critical and non-critical forms of COVID-19 did not differ significantly with regard to the median (interquartile range) TSH level (0.96 (0.68–1.71) vs. 1.27 mIU/L (0.75–1.79), *p* = 0.40). Abnormal TSH level was encountered in 17 patients (12.2%); most of them had subclinical thyroid disease. TSH assay data at the three-month post-discharge visit were available for 151 patients without prior thyroid disease. Only seven of them (4.6%) had abnormal TSH levels. Median TSH level at the post-discharge visit was significantly higher than median TSH level during hospitalization. Conclusions: Our findings suggest that COVID-19 is associated with a transient suppression of TSH in a minority of patients regardless of the clinical form. The higher TSH levels three months after COVID-19 might suggest recovery from non-thyroidal illness syndrome.

## 1. Introduction

Coronavirus disease 19 (COVID-19) has a broad spectrum of clinical manifestations, ranging from mild or asymptomatic disease (in ≈80% of cases) to severe disease requiring hospital admission (in 15 to 29% of cases) and critical disease with acute respiratory distress syndrome, a cytokine storm, and multi-organ failure requiring intensive care unit (ICU) admission (in 5 to 9% of cases) [1,2]. Hypoxia and elevated levels of proinflammatory cytokines are prominent features of severe and critical forms of COVID-19. Hypoxia and elevated levels of proinflammatory cytokines are also known to be associated with non-thyroidal illness syndrome (NTIS) [3,4], a term that refers to changes in thyroid function tests occurring in patients with critical illness requiring ICU admission [5,6]. One can therefore reasonably expect critical COVID-19 to be associated with an authentic NTIS.

In the context of COVID-19, it is important to bring to mind that the TSH-positive cell count and the TSH immunostaining intensity were lower in pituitary samples from autopsies of patients with the previous severe acute respiratory syndrome coronavirus (SARS-CoV) when compared to normal pituitary samples [7] and that the thyroid gland samples showed destruction of the follicular epithelium and exfoliation of epithelial cells into the follicles [8,9]. Moreover, one report in Chinese described significantly lower serum levels of T3, T4, and TSH in patients infected with SARS-CoV than in control subjects and the decrease in free T3 (fT3) levels was more pronounced in severe forms of the disease [10]. Biological hypothyroidism was also reported in 4 out of 61 SARS-CoV survivors who were evaluated for hormonal derangements three months following recovery [11]. SARS-CoV-2, therefore, may affect the hypothalamic-pituitary-thyroid axis leading to perturbations in thyroid function tests [12,13]. Indeed, pathological changes (degeneration, necrosis, etc.) were observed in thyroid specimens obtained from three patients who died from COVID-19 [14]. Moreover, the SARS-CoV-2 receptor for viral entry (angiotensin-converting enzyme 2 (ACE2)) and the transmembrane protease serine 2 (TMPRSS2), which is required for priming of the virus’s spike (S) protein, are expressed in the thyroid [15,16,17,18].

Alterations in thyroid function tests in patients hospitalized with COVID-19 may also arise from the side effects of iodine contrast media which could cause iodine-induced hyperthyroidism (i.e., the Jöd-Basedow phenomenon) when the normal response to excess iodine (the acute Wolff–Chaikoff effect) is impaired [19] or from treatment with corticosteroids which are well known to suppress the TSH release [20]. Of note, these two possibilities are not specific to COVID-19 but would apply to any patient receiving iodine contrast media or corticosteroids.

Hence, we performed an observational study of TSH levels among patients admitted to a university hospital with COVID-19. We compared TSH levels between patients with critical COVID-19 (corresponding to ICU admission or death) and those with non-critical COVID-19, i.e., survivors requiring hospitalization but not ICU admission. Thereafter, thyroid function tests were evaluated in patients three-months post-discharge to look at the thyroid status in the medium term after SARS-CoV-2 infection.

## 2. Patients, Materials, and Methods

### 2.1. Study Population and Data Collection

We retrospectively extracted data from the electronic medical records for consecutive adult patients hospitalized with laboratory-confirmed COVID-19 at Amiens University Hospital (Amiens, France) between the start of the COVID-19 epidemic in our region (27 February 2020) and 29 April 2020. Confirmed COVID-19 was defined as a nasopharyngeal swab specimen that had tested positive in a reverse-transcriptase polymerase chain reaction assay. The main inclusion criteria were a confirmed diagnosis of COVID-19 and inpatient admission to Amiens University Hospital. The main exclusion criteria were opposition to data collection by the patient or his/her legal guardian and age under 18. Collected data included demographics, cardiovascular risk factors, medical history, medications of special concern (levothyroxine, corticosteroids, amiodarone, antithyroid drugs, and immunotherapy), the main clinical variables, routine laboratory results, thyroid function tests, administration of iodine contrast media and corticosteroids, and the main clinical outcomes.

After hospital discharge, all survivors were invited to return for a three-month post-discharge visit regardless of the availability of TSH assay data at baseline. During this outpatient visit relevant clinical events since hospital discharge were recorded, a clinical examination was performed and a prescription for thyroid function tests was delivered.

The study was designed in accordance with the declaration of Helsinki and conducted in accordance with good clinical practice guidelines and French legislation on clinical research and data protection. In line with the French legislation on analysis of data gathered during routine clinical practice, the first part of the study (data collected during the hospitalization period) was approved by the institutional review board at Amiens University Hospital and registered with the French National Data Protection Commission (CNIL: Commission nationale de l’informatique et des libertés (Paris, France)); reference: PI2020_843_0051. Patients who expressed their opposition to data collection were excluded from the study. Concerning the second part of the study (data collected at the post-discharge visit), it was approved by an independent ethics committee and registered with French National Data Protection Commission and with the national agency for the safety of medicines and health products (ANSM: Agence nationale de sécurité du médicament et des produits de santé); reference: ID-RCB 2020-A02129-30. It was also registered to ClinicalTrials.gov (Identifier: NCT04563676). All patients gave informed consent to participate.

All data were anonymized upon extraction, and only anonymized data were analyzed. All data were double-checked by the first author, who vouches for the data’s accuracy.

### 2.2. Study Groups and Outcomes

The M0 group comprised all patients with available TSH data during the acute phase (hospitalization) who had no history of thyroid disease. The M3 group comprised all patients with available thyroid function tests three-months post-discharge and who had no history of thyroid disease.

Firstly, TSH levels in the M0 group were compared between patients with critical form (i.e., those resulting in inpatient death or ICU admission) and patients with non-critical form of COVID-19 (i.e., those requiring hospitalization but not ICU admission, and which did not result in death (the patient was discharged alive)). The proportions of patients with low or high TSH levels were then compared between patients with critical form and those with non-critical form. Moreover, free T4 (fT4) and fT3 levels, when available, were also compared between patients with critical form and those with non-critical form.

At the three-month post-discharge visit, thyroid function tests were summarized descriptively to look at the thyroid status in the medium term after SARS-CoV-2 infection. Additionally, TSH levels at the post-discharge visit were compared to those performed during the acute phase in a subset of patients who had TSH assay data at both time points.

### 2.3. Materials

During the acute phase (hospitalization), all samples were analyzed at Amiens University Hospital’s laboratory. Serum levels of TSH and thyroid hormones (fT4 and fT3) were analyzed using a chemiluminescence immunoassay system (Atellica^®^, Siemens Healthineers, Erlangen, Germany). Normal levels were 0.4–4.0 mIU/L, 11.5–22.7 pmol/L (0.89–1.76 ng/dL), and 3.5–6.5 pmol/L for TSH, fT4, and fT3, respectively.

Concerning the three-month post-discharge data, thyroid function tests were performed at different laboratories. Normal ranges were not the same between those different laboratories. We, therefore, recorded the values and specified whether they were normal, low, or high according to the laboratory sheet from the laboratory where the sample was analyzed.

### 2.4. Statistical Analysis

Baseline demographics and clinical characteristics were expressed as the median (interquartile range (IQR)) for numerical variables and the frequency (percentage) for categorical variables.

Comparisons between clinical forms of COVID-19 were performed with the Mann–Whitney–Wilcoxon test for numerical variables and Fisher’s exact test for categorical variables. To assess the correlation between thyroid function tests and inflammatory markers such as the CRP levels the Spearman’s rank correlation test was used.

TSH levels at the post-discharge visit were compared with TSH levels during hospitalization using the paired Wilcoxon signed-rank test in the subset of patients who had TSH data available at both time points.

All statistical tests were two-sided and were performed with R software (version 4.0.0, R Core Team, R Foundation for Statistical Computing, Vienna, Austria). The threshold for statistical significance was set to *p* < 0.05.

## 3. Results

A total of 448 patients were admitted to Amiens University Hospital between the start of the COVID-19 outbreak and 29 April 2020. None of the patients opposed data collection. TSH levels during the hospitalization period were available for 191 (42.6%) of the 448 patients (Figure 1). The baseline characteristics of patients with TSH assay data during hospitalization and those without are summarized in Table 1. When compared with the group lacking TSH data, the group with TSH data had a higher median age, a lower median BMI, a higher proportion of women, and higher proportions of patients with arterial hypertension and thyroid diseases; all *p* < 0.05. Among the 191 patients with available TSH data during hospitalization, 52 had a history of thyroid diseases and were excluded from subsequent analyses, while the remaining 139 constituted the M0 group.

### 3.1. TSH Levels in the Acute Phase (the M0 Group)

Among the 139 patients with available TSH data during hospitalization and no history of thyroid disease, the median TSH level was 1.12 (0.71–1.76) mIU/L and the median time interval between hospital admission and blood sample collection for a TSH assay was 3 (2–8) days. Patients with critical and non-critical forms of COVID-19 did not differ significantly with regard to the median (IQR) TSH level (0.96 (0.68–1.71) vs. 1.27 mIU/L (0.75–1.79), respectively; *p* = 0.40) (Figure 2). Low TSH levels (<0.4 mIU/L) were encountered in fourteen patients: six patients (10.7%) in those with critical form vs. eight (9.6%) in those with non-critical form, *p* = 1. Nine of the fourteen patients had available fT4 assay, none of them had high fT4 levels but two had low fT4 suggesting central hypothyroidism or NTIS (Table 2). High TSH levels (>4 mIU/L) were encountered in three patients: three patients (5.4%) in those with critical form vs. 0 (0%) in those with non-critical form, *p* = 0.063 (Table 2). Two of the three patients had low fT4 levels suggesting primary hypothyroidism. Of note, the median (IQR) time interval between hospital admission and blood sample collection for a TSH assay was 4 days (2–12.5) in those with critical form vs. 3 days (1–6) in those with non-critical form; *p* = 0.011.

Levels of free T4 and free T3 were measured in a minority of patients in the M0 group (N = 34). In those patients with available thyroid hormones levels, the median (IQR) fT4 level was 13.0 (10.4–15.3) pmol/L in patients with critical form vs. 16.6 (15.0–17.2) in those with non-critical form, *p* = 0.0123. The median (IQR) fT3 level was 3.09 (2.57–3.71) pmol/L in patients with critical form vs. 4.13 (3.18–4.58) in those with non-critical form, *p* = 0.0192. Among the 34 patients with fT3 measurement, fT3 were low (<3.5 pmol/L) in 19 patients (14 in patients with critical form and 5 in patients with non-critical form).

It is worth mentioning that there was no significant correlation between TSH levels and CRP levels on admission. However, there was a significant negative correlation between fT3 levels and CRP levels (correlation coefficient −0.3437049, *p* = 0.0223).

### 3.2. TSH Levels at the Three-Month Post-Discharge Visit (the M3 Group)

Among the patients who were discharged alive, 209 had returned for a post-discharge visit by 1 March 2021. TSH assay was performed in 173 patients (Figure 1) at a median interval between hospital admission and blood sample collection for a TSH assay of 122 (101–148) days. Among the 173 patients with available TSH assay data at the post-discharge visit, 22 had a history of thyroid diseases and were excluded from subsequent analyses and the remaining 151 constituted the M3 group.

The median TSH level at the post-discharge visit was 1.66 (1.28–2.19) mIU/L. Only three patients had low TSH levels at the post-discharge visit (2.0%); fT4 levels were normal in two of them while the third patient had low fT4 compatible with central hypothyroidism (due to pituitary apoplexy in the aftermath of COVID-19) (Table 3). High TSH levels at the post-discharge visit were encountered in four patients (2.6%); none of them had abnormal fT4 levels suggesting that they all have subclinical hypothyroidism (Table 3).

In the subset of patients who have TSH assay data available at both time points (during hospitalization and at the post-discharge visit, *n* = 53), the median TSH level (IQR) at the three-month post-discharge visit was 1.50 (1.21–2.00) vs. 0.94 mIU/L (0.64–1.85) at the acute phase, *p* = 0.0003 (paired Wilcoxon test) (Figure 3).

## 4. Discussion

The main findings of our study are as follows. Firstly, during the acute phase, abnormal TSH levels were present in 12.2% of inpatients with COVID-19 and available TSH data; two patients (1.4%) had central hypothyroidism or NTIS and two patients (1.4%) had primary hypothyroidism while the remaining patients (9.4%) had subclinical thyroid disease. Secondly, we did not observe a statistically significant difference in TSH levels between inpatients with critical COVID-19 and those with a non-critical form of the disease whereas we did observe statistically different fT3 and fT4 levels between the two forms, albeit in a small number of patients (<25% of patients with TSH data). Thirdly, TSH levels were in the normal range in 95% of patients at the post-discharge visit, only one patient had central hypothyroidism whereas the remaining patients had subclinical thyroid disease only. Finally, we observed a statistically significant difference in TSH levels between the acute phase and the post-discharge visit with a shift toward higher TSH levels three-months post-COVID-19; this higher TSH levels three months after COVID-19 might suggest recovery from NTIS.

Since the start of the COVID-19 pandemic in December 2019, many observational studies looked at the impact of this infection on the thyroid gland. Our findings about the prevalence of abnormal TSH levels in inpatients with COVID-19 are in line with those from most studies that have reported thyroid function tests in the acute phase of COVID-19: Gao et al. 15% [21], Lui et al. 6.3% [22], Khoo et al. 11.1% [23], Zheng et al. 9.8% [24] and Ahn et al. 17.6% [25]. However, some studies reported a higher rate of abnormal TSH levels in inpatients with COVID-19. In one study from China, lower than normal TSH levels were observed in 28 out of 50 patients with moderate to critical COVID-19 [26]. The median (IQR) TSH level was 0.30 mIU/L (0.15–0.86) in this Chinese study vs. 1.12 (0.71–1.76) in the present study. Baseline characteristics of the Chinese study population were not available from the publication, but the mean age was ≈48 years, which is considerably lower than the value for the patients of our study. Moreover, most patients (31/50) were receiving corticosteroids when thyroid function tests were assessed and information about the use of iodine-contrast media was absent. In another study from Italy, abnormal TSH levels were observed in 73 out of 287 patients (25.4%) hospitalized for COVID-19 in non-intensive care units. Thyroid dysfunction was overt in 33 cases (31 thyrotoxicosis and 2 hypothyroidism) and subclinical in the remaining 40 patients [27]. In seven patients with thyrotoxicosis who were followed up, thyroid function tests namely fT4 improved rapidly (5–10 days) regardless of the treatment of thyrotoxicosis. It is therefore possible that the time point of TSH measurement is a key factor. Thyroid function tests were performed on the first day of hospitalization in the Italian study versus a median of 3 days in the present study. Interestingly, Świątkowska-Stodulska et al. assessed thyroid function tests in patients admitted to the hospital for COVID-19 at three different time points. They found that abnormal thyroid function tests were present in 57.5% of the patients on day 1, 63.1% on day 4, and 76.6% on day 10 [28].

Although the TSH levels among patients with critical form were numerically lower in comparison with the TSH levels in patients with non-critical form, the difference was not statistically significant. Muller et al. compared thyroid function tests between 85 patients admitted to high intensive care units (HICU-20) because of COVID-19 and 41 patients admitted to low-intensity care units (LICU-20) for the same reason. The median (IQR) TSH level was 1.04 mIU/L (0.47–1.80) in the HICU-20 group vs. 1.43 (0.71–2.28) in the LICU-20 group and the difference between the two groups was statistically significant. Moreover, low TSH levels (<0.45 mIU/L) were present in 24.7% of patients in the HICU-20 group vs. 9.8% in the LICU-20 group, *p* < 0.05 whereas high TSH levels were present in 3.5% and 9.8% in the HICU-20 and LICU-20 groups, respectively [29]. The authors of this study speculated that higher CRP levels could mean a greater systemic spread of SARS-CoV-2 impacting the thyroid gland. It is possible that the absence of difference in terms of TSH levels between critical and non-critical forms of COVID-19 resulted from an insufficient number of patients in our study. However, some studies found a difference in TSH levels between moderate and severe/critical COVID-19 even if the number of patients was low [21,25,30,31,32,33,34] whereas others did not find a difference in TSH levels despite the higher number of patients [35]. It is also possible that factors other than systemic inflammation markers (i.e., CRP levels) participate in thyroid dysfunction in patients with COVID-19 such as: treatment with corticosteroids [28], use of iodinated contrast media, the background population iodine intake and the existence of iodine deficiency. Finally, in agreement with the reported data [25,31,33,36] levels of fT3 and fT4 were lowers in patients with critical COVID-19 as compared with those with non-critical form. Of note, these alterations in thyroid function among inpatients with COVID-19 are not innocuous since they were associated with poor prognosis in retrospective studies [37,38,39].

Our study found that thyroid dysfunction is rare three months after COVID-19. This finding is supported by the findings from the above-mentioned Chinese [21,22,26] and Italian studies [27,29]. Moreover, even though many cases of subacute thyroiditis during the post-COVID-19 period were described [12,13], thyroid function tests were normalized within six weeks in almost all cases. Furthermore, in a study from the United Kingdom, 55 patients had follow-up thyroid function tests at a median of 79 days after admission to the hospital with COVID-19, 47 of them were euthyroid and 4 had central hypothyroidism whereas the remaining 4 had subclinical thyroid diseases [23]. In another study from the United Kingdom, thyroid function tests were normal in all patients attending a follow-up visit ≥3 months after COVID-19 (*n* = 68) [40]. All these data indicate that thyroid function abnormalities in the context of COVID-19 disappear or greatly improve in the medium-term [41,42,43,44]. However, long-term follow-up data are warranted since genetic liability to COVID-19 was associated with hypothyroidism [45]. Moreover, incident anti-thyroid peroxidase (anti-TPO) positivity was observed during follow-up in one study [41].

Interestingly, TSH levels three months after COVID-19 were significantly higher than TSH levels in the acute phase (during hospitalization), which indirectly suggest that TSH levels were slightly suppressed in the acute phase. An interesting study from the United Kingdom compared TSH levels before, during, and after COVID-19 and showed that patients with COVID-19 had lower TSH levels (1.02 mIU/L (0.60–1.65)) when compared with their earlier (before COVID-19) TSH levels (1.59 (1.03–2.24) in 2019) suggesting that the TSH is slightly suppressed in the acute phase of COVID-19. TSH levels returned to baseline at follow-up suggesting that thyroid function alterations in patients with COVID-19 show rapid recovery [23]. These results about suppressed TSH levels in the acute phase of COVID-19 are supported by a recent study that showed suppressed mRNA of pituitary hormones and pituitary regulatory genes in lethal COVID-19 cases (*n* = 23) [46].

Our study had some limitations. Firstly, comprehensive thyroid function test data were not available for all study participants during hospitalization. However, our study was observational, and each patient’s attending physician chose whether or not to prescribe assays for TSH and/or other hormones. Secondly, we did not have data about some relevant inflammatory markers such as interleukin-6. TSH levels in patients hospitalized for COVID-19 were correlated with IL-6 levels in one study [27]. Thirdly, we did not perform a thyroid ultrasound to detect morphologic changes in the thyroid gland or thyroid scintigraphy to search for thyroid uptake abnormalities. In a small study from Italy, thyroid function tests returned to normal in 28 out of 29 patients with COVID-19-related thyrotoxicosis but hypoechogenicity on thyroid ultrasound persisted in 10 of the patients (34.5%) [42]. Fourthly, no control group of hospitalized patients without COVID-19 was included. Lastly, the observational design prevented us from prespecifying the statistical power, so our results should be considered as being indicative only.

## 5. Conclusions

In conclusion, the results of this observational study of 139 inpatients with COVID-19 and TSH assay data showed that TSH levels did not differ according to whether or not the disease was critical (i.e., having led to ICU admission or death). The post-discharge data of 151 patients showed that thyroid function tests were normal in 95% of patients three months after the infection with SARS-CoV-2 suggesting that thyroid dysfunction in patients with COVID-19 has short-duration and self-limited course. The comparison between TSH levels during hospitalization and those performed three months after COVID-19 enabled us to detect a slightly suppressed TSH level in the acute phase that was not visible when considering data from the acute phase only.

## Figures and Tables

**Figure 1 life-12-02014-f001:**
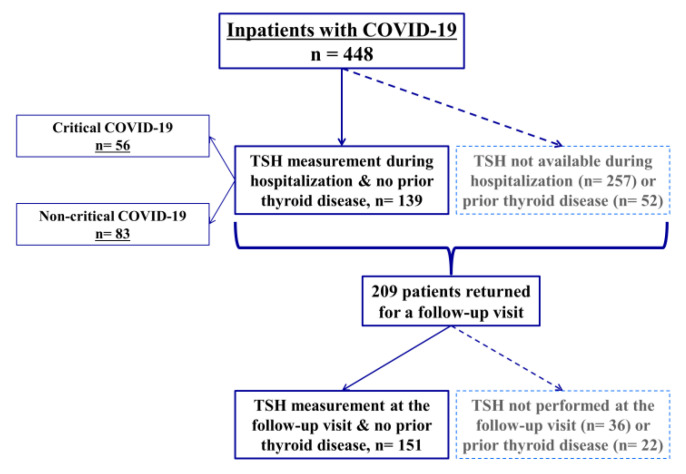
Study flow chart showing the numbers of patients hospitalized with COVID-19 (*n* = 448), those with TSH assay data available during hospitalization (with and without critical form of COVID-19), and those who returned for a three-month post-discharge visit. The number of patients without history of thyroid disease at each step is also shown (*n* = 139 during hospitalization and *n* = 151 at the post-discharge visit). A total of 53 patients had TSH assay data available at both time points (not shown in the flow chart).

**Figure 2 life-12-02014-f002:**
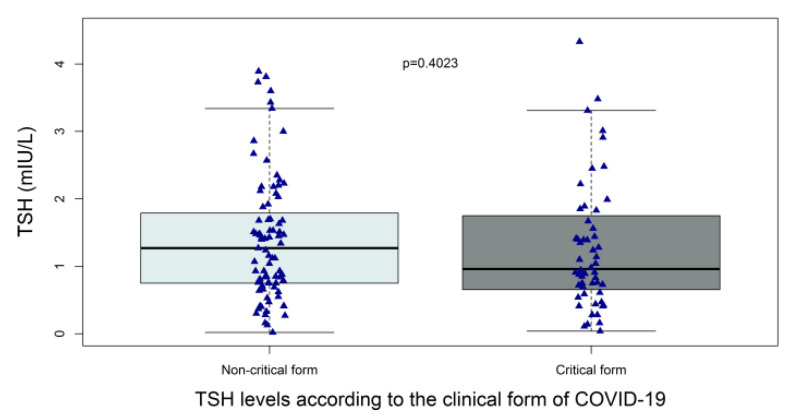
Distribution of TSH levels in hospitalized patients with and without critical forms of COVID-19. Patients with available TSH assay data in the acute phase and without prior thyroid disease (N = 139). TSH levels plotted with box indicating 25th and 75th percentiles, whiskers indicating 5th and 95th percentiles, line in box indicates median. *p* values are from Mann–Whitney–Wilcoxon test. Of note, two patients with critical form who had TSH levels of 7.3 and 15 mIU/L are not represented in the Figure (outliers).

**Figure 3 life-12-02014-f003:**
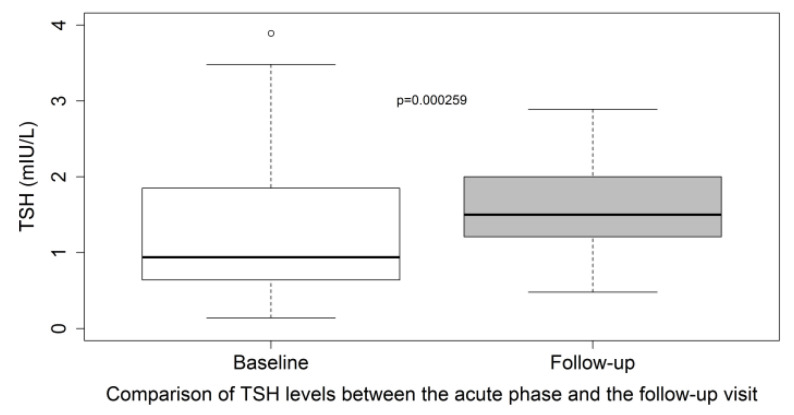
Comparison of TSH levels between the acute phase and the post-discharge visit in the subset of patients with TSH assay data available at both time points (*n* = 53). TSH levels plotted with box indicating 25th and 75th percentiles, whiskers indicating 5th and 95th percentiles, lines in boxes indicate median. *p* values are from the paired Wilcoxon signed-rank test.

**Table 1 life-12-02014-t001:** Baseline characteristics of patients with and without TSH assay data at the acute phase (during hospitalization).

Characteristic		TSH Assay Data not Available at M0 (N = 257)		TSH Assay Data Available at M0 (N = 191)		*p* Value
		Median (IQR) or *n* (%)	Missing Data	Median (IQR) or *n* (%)	Missing Data	
**Demographic data**						
**Age (year)**		69 (56–82)	0	78 (66–86)	0	**<0.0001**
**Men**		155 (60.3%)	0	89 (46.6%)	0	**0.0041**
**Body mass index (kg/m²)**		29.0 (24.1–32.9)	62	26.5 (22.5–31.2)	38	**0.0054**
**Smoking history**			77		64	0.593
	Never	112 (43.6%)		84 (44.0%)		
	Former smoker	59 (23.0%)		35 (18.3%)		
	Current smoker	9 (3.5%)		8 (4.2%)		
**Coexisting conditions**						
**Obesity**		91 (35.4%)	56	52 (27.2%)	34	**0.0225**
**Diabetes**		66 (25.7%)	0	61 (31.9%)	0	0.168
**Arterial hypertension**		144 (56.0%)	0	127 (66.5%)	0	**0.0314**
**Hyperlipidemia**		81 (31.5%)	1	72 (37.7%)	0	0.191
**Cardiovascular diseases**		134 (42.4%)	0	62 (45.5%)	0	0.563
**Pulmonary Disease**		39 (15.2%)	0	23 (12.0%)		0.407
**Chronic kidney disease**		37 (14.4%)	0	31 (16.2%)	0	0.597
**Cancer**		41 (15.9%)	0	33 (17.3%)	0	0.797
**Thyroid disease**		18 (7.0%)	0	52 (27.2%)	0	**<0.0001**
**Main laboratory findings on admission**						
**Glycaemia (mmol/L)**		6.7 (5.8–7.9)	27	6.5 (5.6–8.5)	12	0.691
**GFR (MDRD mL/min/1.73 m²)**		81.5 (56–110)	5	79 (55–99)	0	0.507
**ALAT >40 U/L**		81 (31.5%)	32	51 (26.7%)	11	0.110
**ASAT > 40 U/L**		127 (49.4%)	31	93 (48.7%)	11	0.367
**GGT > ULN**		107 (41.6%)	42	84 (44%)	17	0.838
**WBC × 10^9^ per L**		6.4 (5–9.1)	6	6.5 (4.6–9.2)	0	0.868
**Lymphocytes × 10^9^ per L**		0.9 (0.6–1.2)	6	0.8 (0.6–1.2)	0	0.340
**CRP mg/L**		89.2 (42.0–152.1)	7	82 (25.3–160.6)	0	0.332

Abbreviations: ALAT, alanine transaminase; ASAT, aspartate transaminase; CRP, C-reactive protein; GFR (MDRD), glomerular filtration rate as estimated by the Modification of Diet in Renal Disease equation; GGT, gamma-glutamyltranspeptidase; IQR, interquartile range; N, number; ULN, upper limit of normal; WBC, white blood cells.

**Table 2 life-12-02014-t002:** Main characteristics and thyroid hormones levels of patients with abnormal TSH level during the acute phase of COVID-19 (hospitalization).

Patient N°	1	2	3	4	5	6	7	8	9	10	11	12	13	14	15	16	17
Age (years)	51	81	69	71	70	80	64	89	67	88	56	29	92	45	72	59	65
Sex (M = Male, W = female)	M	M	W	W	M	W	M	W	M	W	M	M	W	W	M	W	M
TSH (N: 0.4–4 mIU/L)	**0.33**	**0.28**	**0.13**	**0.28**	**0.37**	**0.3**	**0.16**	**0.27**	**0.04**	**0.16**	**0.11**	**0.14**	**0.02**	**0.28**	**7.3**	**4.3**	**15.0**
fT4 (N: 11.5–22.7 pmol/L)	NA	NA	15.9	16	16.9	17.2	**8.5**	NA	NA	17.9	11.6	**10.7**	NA	15.5	**10.1**	NA	**9.4**
fT3 (N: 3.5–6.5 pmol/L)	NA	NA	4.6	3.8	**2.94**	NA	**2.7**	NA	NA	**3.3**	**2.8**	**3.4**	4.6	**3.0**	3.6	NA	4.1
Critical form of COVID-19	No	No	No	Yes	No	No	Yes	No	Yes	No	Yes	Yes	No	Yes	Yes	Yes	Yes

**Table 3 life-12-02014-t003:** Main characteristics and thyroid hormone levels of patients with abnormal TSH levels at the three-month post-discharge visit.

Patient N°	I	II	III	IV	V	VI	VII
Age (years)	43	69	67	51	56	38	77
Sex (M = Male, W = female)	W	W	M	W	W	M	W
TSH (mIU/L)	**0.31**	**0.36**	**0.27**	**4.5**	**5.8**	**7.5**	**10.4**
fT4 (pmol/L)	13.6	**1.5**	12.9	13.3	14.9	15.7	15.5
fT4 within normal limits	Yes	**Low**	Yes	Yes	Yes	Yes	Yes
fT3 (pmol/L)	NA	NA	NA	NA	NA	NA	NA
Interval between admission and TSH assay (days)	147	99	167	84	144	95	129
TSH during hospitalization	NA	NA	NA	NA	NA	NA	NA

## Data Availability

Data are not available due to French legislation.

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
