# Peer review of "Thyrotropin Levels in Patients with Coronavirus Disease 2019: Assessment during Hospitalization and in the Medium Term after Discharge"

_life, 2022, doi:10.3390/life12122014_

Round 1

Reviewer 1 Report

Al-Salameh et al in their study titled “Thyrotropin levels in patients with coronavirus disease 2019: assessment during hospitalization and in the medium term after discharge” compared TSH levels between inpatients with critical versus non-critical coronavCOVID-19 and described the status of TSH levels three months after hospitalization. Generally, the ensemble of non thyroidal illness in the critically ill patients is well documented in literature.

The paper is an interesting read and well written with good supporting literature. Where it lacks is in the methodology and the results section and thereby the interpretation, which are the areas of major concern. The uniformity of the number of patients studies was also extremely confusing and calls into question the reliability.

In the abstract the authors mention that the sample size was 139 patients who had a measured TSH level, and then mention that the patients were seen at the follow up, which creates a picture that these patients were the same.

 In the study methodology the authors have to mention clearly that this was a retrospective study and the time frame in which it was carried out rather than mention “between the start of the COVID-19 epidemic in our region and April 29th, 2020”. Although a flow chart is provided for the study work flow, the number of patients mentioned in the manuscript does not corroborate with the diagram. Moreover the actual number of patients from the M0 group and present in the M3 group is unclear. How many patients from the 139 were included among the 209 patients? Without knowing this, the validity of the statistical analysis using “paired Wilcoxon signed-rank test in the subset of patients who had TSH data available at both time points” cannot be judged and neither can Figure 3 be interpreted.

Where is the M1 and M2 group.

The authors mentioned that “Collected data included demographics, cardiovascular risk factors, medical history, medications of special con-cern (levothyroxine, corticosteroids, amiodarone, antithyroid drugs, and immunothera-py), the main clinical variables, routine laboratory results, thyroid function tests, administration of iodine contrast media and corticosteroids, and the main clinical outcomes” but no data was provided for the same.

Moreover, the description in section 3.1 is very confusing and does not correspond with the Figure 2 and this is a major concern on the interpretation of the results. The authors need to check the veracity of their data. Its not clear what the 17 number of patients in Table 1 represent?

It is not clear whether any of the medications that the patients received in the ICU impacted the levels of the hormones. The past history and the presence of any other comorbidities or prior illnesses has not been mentioned.

Keeping these points in mind, the interpretation and conclusion of the authors cannot be accepted.

Reviewer 2 Report

The study investigates the TSH levels between inpatients with critical versus non-critical coronavirus disease 19 (COVID-19), 2) and the status of TSH levels three months after hospitalization.

The major concern is why check TSH level only? How about the other important thyroid function indicators such as T3, T4, TSH-R Ab, anti- TPO, anti- TG…. The TSH level only cannot refer to any important clinical significance.

TSH assay data during hospitalisation were available for “139” patients without prior thyroid disease. However, TSH assay data at the 3-month follow-up visit were available for “151” patients without prior thyroid disease. Why follow-up data in 151 patients more than prior 139 patients?

Reviewer 3 Report

Dear authors,

The manuscript is well written and with great value for the readers. I only have a few suggestions:

-please revise the English

-add a Limitation subsection

-put the subtitles in section 2 with the specific style

-if you could add some more references, it would be great for the paper

Round 2

Reviewer 1 Report

The authors have tried to answer the questions raised in the first round of review, but yet there are few more concerns that have not been addressed.

As they have rightly mentioned, the study had two arms, the retrospective and prospective. The patients that were seen in the retrospective arm were not the same in the prospective arm except for 53 patients. The authors say that  ‘ we performed a prospective evaluation of thyroid function tests three months following hospital discharge to look on the thyroid status in the medium term after SARS-CoV-2 infection’ implies that the same patients were prospectively followed in time. Moreover, the term ‘follow up’ is misleading as it implies that the same patients were followed up. At best the study compares between the TSH levels among hospitalised patients and those that have recovered from COVID after 3 months. It would be better, in my opinion, to say that ‘ levels of TSH were evaluated in patients 3 months post recovery’.

It is always nice to define the study groups in the methodology section. In section 2.2 the authors mentioned ‘The M0 group comprised all patients with available TSH data during the acute phase (hospitalization) and no history of thyroid disease. The M3 group comprised all patients with available thyroid function tests at the follow-up visit and no history of thyroid disease.’ The point for the M1 and M3 was raised to point out that it is better to define the groups in the methodology for the ease of the readers rather than to let them assume the meaning of the naming of the groups. The description of M3 has to be made clear i.e., 3 months post discharge or 3 months after recovery as only 53 patients can be considered follow up.

. The authors mention that 3 patients ‘High TSH levels (>4 mIU/L) were encountered in three patients: three patients (5.4%) in those with critical form’. In figure 2 the scatter plot shows only 1 data point, were the other 2 patients excluded as outliers?

The conclusion also need to be revised. The authors mention that ‘The prospective follow-up data of 151 patients showed the thyroid function tests were normal’. Only 53 patients in the 151 can be considered as follow up as they had a prior TSH level, while the data for the rest is a single time point measurement.

Reviewer 2 Report

no more comment

Author Response

We would like to thank the Reviewer for taking the time to review our manuscript.

Reviewer 3 Report

The authors have improved the paper. I agree with publication.

Author Response

We would like to thank the Reviewer for taking the time to review our manuscript and for his positive final decision